# Mitigating the Curse of Dimensionality for Certified Robustness via Dual Randomized Smoothing

**Song Xia**    **Yi Yu**    **Xudong Jiang** [*]    **Henghui Ding** [✉]

Nanyang Technological University, Singapore

{xias0002,yuyi0010,exdjiang}@ntu.edu.sg, henghui.ding@gmail.com

## Abstract

Randomized Smoothing (RS) has been proven a promising method for endowing an arbitrary image classifier with certified robustness. However, the substantial uncertainty inherent in the high-dimensional isotropic Gaussian noise imposes the curse of dimensionality on RS. Specifically, the upper bound of $\ell_2$ certified robustness radius provided by RS exhibits a diminishing trend with the expansion of the input dimension $d$, proportionally decreasing at a rate of $1/\sqrt{d}$. This paper explores the feasibility of providing $\ell_2$ certified robustness for high-dimensional input through the utilization of dual smoothing in the lower-dimensional space. The proposed Dual Randomized Smoothing (DRS) down-samples the input image into two sub-images and smooths the two sub-images in lower dimensions. Theoretically, we prove that DRS guarantees a tight $\ell_2$ certified robustness radius for the original input and reveal that DRS attains a superior upper bound on the $\ell_2$ robustness radius, which decreases proportionally at a rate of $(1/\sqrt{m} + 1/\sqrt{n})$ with $m + n = d$. Extensive experiments demonstrate the generalizability and effectiveness of DRS, which exhibits a notable capability to integrate with established methodologies, yielding substantial improvements in both accuracy and $\ell_2$ certified robustness baselines of RS on the CIFAR-10 and ImageNet datasets. Code is available at https://github.com/xiasong0501/DRS.

## 1 Introduction

Deep neural networks have shown great potential for an ever-increasing range of complex applications (Yu et al., 2023; Wu et al., 2024; Ding et al., 2023b;a;c; Ju et al., 2023; Luo et al., 2023; Kong et al., 2022). While those models achieve commendable performance and manifest a capacity for generalization within the distribution they are trained on, they concurrently exhibit a pronounced vulnerability to adversarial examples (Biggio et al., 2013; Szegedy et al., 2014; Tramer et al., 2020; Yu et al., 2022). By adding a small and human imperceptible perturbation to the image, the adversarial examples could mislead the well-trained deep neural network at a high success rate.

In response to these adversarial vulnerabilities, researchers have explored a variety of methods to enhance the robustness of deep neural networks. The empirical defense method, *e.g.*, adversarial training (Madry et al., 2018; Ding et al., 2019; Shafahi et al., 2019; Sriramanan et al., 2021; Cheng et al., 2023), bolsters model robustness through iterative training with adversarially generated examples. However, this kind of empirical defense is not fully trustworthy. Many empirical defense approaches are later broken by more intricately crafted adversarial attacks (Carlini & Wagner, 2017; Yuan et al., 2021; Hendrycks et al., 2021; Duan et al., 2021; Li et al., 2023). This inspires researchers to develop methodologies capable of providing certified robustness, *e.g.*, guaranteeing the classifier to return a constant prediction result within a certain range (Raghunathan et al., 2018; Wong & Kolter, 2018; Hao et al., 2022; Kakizaki et al., 2023).

Randomized Smoothing (RS) has been proven a promising method to provide certified robustness for any kind of classifiers. Initially introduced by Lecuyer et al. (2019), RS is guaranteed with a loose robustness boundary, and later Cohen et al. (2019) theoretically proves a tight robustness boundary

---

[*] Research lead and supervisor
[✉] Corresponding author (henghui.ding@gmail.com)

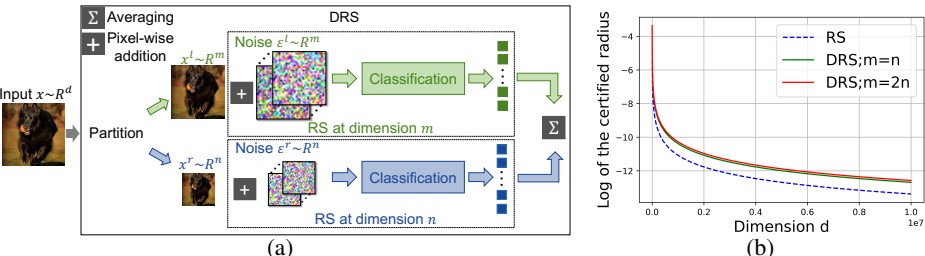

Figure 1: (a) The smoothing process of DRS. (b) The upper bound of $\ell_2$ certified radius (calculated by Equation 4 and Equation 15) of RS and DRS with $\sigma = 1/\sqrt{d}$ and smoothed probability $= 0.999$.

via the Neyman-Pearson lemma (Neyman & Pearson, 1933). In Cohen et al. (2019), by corrupting the original image with the isotropic Gaussian noise at the same dimension, RS turns any base classifier into the smoothed one by predicting with the "majority vote" among the noised samples. The smoothed classifier is thereby assured to exhibit certified robustness to inputs characterized by confident classifications. Specifically, it has been substantiated that the smoothed classifier consistently produces an unchanging classification outcome within a predetermined range, typically a $\ell_2$ or $\ell_\infty$ ball centered around the original input. Meanwhile, RS does not necessitate any a prior assumptions regarding the parameters or architectural configurations of classifiers, making it feasible to provide certified robustness for most deep neural network models.

However, the high uncertainty caused by Gaussian noise not only decays the classification accuracy but also imposes the curse of dimensionality (Kumar et al., 2020; Wu et al., 2021), which makes the upper bound of $\ell_2$ certified radius provided by RS progressively diminishes at a rate of $1/\sqrt{d}$. The predominant emphasis in existing research focuses on refining RS by training classifiers with enhanced capability of predicting noise-corrupted images, such as training the classifier with the Gaussian noise augmented images (Cohen et al., 2019), utilizing adversarial training (Salman et al., 2019), adding an extra deep learning denoiser (Salman et al., 2020; Carlini et al., 2023)), and using model ensemble (Horváth et al., 2022).

While the aforementioned methods demonstrate notable performance gain by either fortifying the classifier or introducing extra denoising modules, they have not effectively addressed the inherent challenge of the curse of dimensionality arising from high-dimensional Gaussian noise. Consequently, the certified robustness provided by RS still continues to diminish fast with the increase of dimension $d$. To mitigate this, we introduce a novel smoothing mechanism termed Dual Randomized Smoothing (DRS). As illustrated in Figure 1, by partitioning the original $d$-dimensional input into two sub-inputs with lower dimensionality of $m$ and $n$, DRS offers certified robustness for the original input through dual smoothing within the lower-dimensional space, and shows a more promising upper bound of $\ell_2$ certified radius. To minimize the information loss caused by the partition of the input for the classification, this paper harnesses the spatial redundancy inherent in the image and partitions the input through image down-sampling with two predefined indexes. The key contributions of this paper can be summarized as follows:

- We introduce a novel smoothing mechanism called Dual Randomized Smoothing (DRS). Theoretically, we prove its capacity in providing a tight $\ell_2$ certified robustness for the high-dimensional input via dual smoothing in the lower-dimensional space.
- We demonstrate that DRS effectively mitigates the curse of dimensionality. DRS yields a superior upper bound for $\ell_2$ robustness, characterized by a slower rate of diminishment.
- We develop the first implementation of DRS by partitioning the input based on the spatial redundancy of the image. Extensive experiments validate the generalizability and effectiveness of DRS. DRS can adeptly integrate with various existing methods, resulting in substantial enhancements to both the accuracy and the certified robustness baseline of RS.

## 2 RELATED WORK

**Certified defense.** Certified defense aims to guarantee that no adversarial example exists within a neighborhood of the input, often a $\ell_2$ or $\ell_\infty$ ball. Those methods typically are either exact ("complete") or conservative ("sound but incomplete"). Exact methods will report whether there exists or not an adversary near the data point to mislead the classifier, usually by mixed integer linear pro-

gramming (Lomuscio & Maganti, 2017; Tjeng et al., 2018) or satisfiability modulo theories (Katz et al., 2017). However, those methods take a large number of computational resources thus being hard to transfer to large-scale neural networks (Tjeng et al., 2018). Conservative methods also certify if there is an adversary existing, but they might decline to make a certification when the data resides in a safe neighborhood. The advantage of such methods lies in the enhanced flexibility and reduced computational resource requirements (Wong & Kolter, 2018; Wang et al., 2018). However, these methods either require specific network architecture (*e.g.*, ReLu activation or layered feed-forward structure) or extensive customization for new architectures. Furthermore, none of them are shown to be feasible to provide defense for modern machine learning tasks such as ImageNet.

**Randomized smoothing.** Randomized Smoothing (RS) shows attractive proprieties in providing certified robustness for any classifier via noise smoothing. RS is first proposed by Lecuyer et al. (2019), which utilizes inequalities from the differential privacy literature to provide a loose $\ell_2$ or $\ell_1$ robustness guarantee for the smoothed classifier. Later, Cohen et al. (2019) proves the tight robustness boundary for $\ell_2$ norm adversary via the Neyman-Pearson lemma, which guarantees much stronger certified robustness. However, the presence of high-dimensional Gaussian noise inevitably erodes the performance of the model. This phenomenon termed the curse of dimensionality was initially noted by Yang et al. (2020) and Kumar et al. (2020), which highlight that the upper bound of the certified radius of randomized smoothing will diminish as the input dimension increases. Wu et al. (2021) extends this observation to encompass a broader range of smoothing distributions, revealing that the $\ell_2$ certified radius of RS diminishes at a rate proportional to $1/\sqrt{d}$.

The predominant focus of existing research lies in enhancing the perdition of the classifier under Gaussian noise corruption. Cohen et al. (2019) trains the classifier with input images augmented by Gaussian noise, and Salman et al. (2019) further enhances this by adversarial training. Zhai et al. (2019) proposes a loss function aiming to maximize the certified radius, and Jeong & Shin (2020) adds a consistency regularization to facilitate the base classifier in producing more consistent predictions under Gaussian distribution. Alternatively, Salman et al. (2020) utilizes a deep learning-based denoiser to purify the corrupted image before classification, and Horváth et al. (2022) utilizes the model ensemble to reduce the overall prediction variance. Most recently, Carlini et al. (2023) introduces a potent denoiser founded on the diffusion model, representing a substantial advancement in noise mitigation that greatly enhances overall robustness. However, prior works have not adequately tackled the challenge of the curse of dimensionality in the context of high-dimensional inputs. Súkeník et al. (2022) proves that the input-dependent RS also suffers from the curse of dimensionality and Pfrommer et al. (2023) proposes a projected randomized smoothing that enhances the lower bound on the certified volume. Different from previous works, this paper first demonstrates the feasibility of providing $\ell_2$ certified robustness for high-dimensional input via dual smoothing in the lower-dimensional space, thus effectively mitigating the disaster caused by dimension expansion. Meanwhile, we theoretically prove DRS can provide a tight $\ell_2$ certified robustness radius for the original high-dimensional input via the lower dimensional smoothing. Additionally, we show that DRS achieves a more promising $\ell_2$ robustness upper bound that decreases at a rate of $(1/\sqrt{m} + 1/\sqrt{n})$ with $m + n = d$.

## 3 PRELIMINARY

### 3.1 RANDOMIZED SMOOTHING

Consider a $k$ classes classification problem with the input $\boldsymbol{x} \in \mathbb{R}^d$ and the label $y \in \mathcal{Y} = \{c_1, \ldots, c_k\}$. RS first corrupts each input $\boldsymbol{x}$ by adding the isotropic Gaussian noise $\mathcal{N}(\boldsymbol{\varepsilon}; 0, \sigma^2 \boldsymbol{I})$. Then it turns an arbitrary base classifier $f$ into a smoothed version $F$ that possesses $\ell_2$ certified robustness guarantees. The smoothed classifier $F$ returns whichever the class the base classifier $f$ is most likely to return among the distribution $\mathcal{N}(\boldsymbol{x} + \boldsymbol{\varepsilon}; \boldsymbol{x}, \sigma^2 \boldsymbol{I})$, which is defined as:

$$F(\boldsymbol{x}) = \arg\max_{c \in \mathcal{Y}} \mathbb{P}(f(\boldsymbol{x} + \boldsymbol{\varepsilon}) = c). \tag{1}$$

**Theorem 1.** *(From (Cohen et al., 2019)), let $f : \mathbb{R}^d \to \mathcal{Y}$ be any deterministic or random function, and $F$ be the smoothed version defined in Equation 1. Let $c_A$ and $c_B$ be the most probable and runner-up classes returned by $F$ with smoothed probability $p_A$ and $p_B$ respectively. Then $F(\boldsymbol{x} + \boldsymbol{\delta}) = c_A$ establishes for all adversarial perturbations $\boldsymbol{\delta}$, satisfying that $\|\boldsymbol{\delta}\|_2 \leq R'$, where*

$$R' = \tfrac{1}{2}\sigma(\Phi^{-1}(p_A) - \Phi^{-1}(p_B)). \tag{2}$$

In Equation 2, $\Phi$ denotes the Gaussian Cumulative Distribution Function (CDF) and $\Phi^{-1}$ signifies its inverse function. Theorem 1 indicates that the $\ell_2$ certified robustness provided by RS is closely linked to the base classifier's performance on the Gaussian distribution; a more consistent prediction within a given Gaussian distribution will return a stronger certified robustness. The proof of Theorem 1 can be found in the Appendix A.1. It is not clear not how to calculate $p_A$ and $p_B$ exactly if $f$ is a deep neural network. Thus Monte Carlo sampling is used to estimate the smoothed probability. This theorem also establishes when we assign $p_A$ with a lower bound estimation $\underline{p_A}$ and assign $p_B$ with a upper bound estimation with $\overline{p_B} = 1 - \underline{p_A}$. Then the radius $R'$ equals:

$$R' = \sigma \left( \Phi^{-1} \left( \underline{p_A} \right) \right). \tag{3}$$

Equation 3 establishes based on $-\Phi^{-1} \left( 1 - \underline{p_A} \right) = \Phi^{-1} \left( \underline{p_A} \right)$. The smoothed classifier $F$ is guaranteed to return the constant prediction $c_A$ around $\boldsymbol{x}$ within the $\ell_2$ ball of radius $R'$.

## 3.2 CURSE OF THE DIMENSIONALITY

While Randomized smoothing shows the intriguing property of providing certified robustness for arbitrary classifiers, it suffers from the curse of dimensionality due to high-dimensional noise corruption (Kumar et al., 2020). This paper delves into the curse of dimensionality of the $\ell_2$ certified robustness proposed by Wu et al. (2021).

**Proposition 1.** *(From (Wu et al., 2021)), for an arbitrary continuous and origin-symmetric distribution $q$, i.e., $\forall z$, $q(z) = q(-z)$, utilized in smoothing, the certified $\ell_2$ radius provided by randomized smoothing of an arbitrary $d$-dimensional input $\boldsymbol{x}$ is bounded by:*

$$r(\boldsymbol{x}) < \frac{5}{\sqrt{d}} \Psi^{-1} \big( \frac{\max_{c \in \mathcal{Y}} \mathbb{P}_{\boldsymbol{\varepsilon} \sim q}(f(\boldsymbol{x}+\boldsymbol{\varepsilon})=c)}{1-5*10^{-7}}; q \big), \tag{4}$$

where $\Psi(r; q) = \int_{\|\boldsymbol{z}\|_2 < r} q(\boldsymbol{z})d\boldsymbol{z}$ is the probability mass of distribution $q$ inside a $\ell_2$ ball with radius $r$ and the parameter $5 * 10^{-7}$ is derived from a predefined error margin between the estimated probability and true probability over the origin-symmetric distribution. Proposition 1 indicates that the $\ell_2$ certified radius provided by RS is limited by an upper bound with a constant multiplier of $1/\sqrt{d}$ and diminishes with the expansion of $d$. The detailed proof can be found in Wu et al. (2021).

## 4 CERTIFIED ROBUSTNESS VIA DUAL RANDOMIZED SMOOTHING

To mitigate the curse of dimensionality that makes the upper bound of certified robustness diminish for high-dimensional input, we propose a new smoothing mechanism called Dual Randomized Smoothing (DRS). As shown in Figure 1, DRS provides certified robustness for arbitrary input $\boldsymbol{x}$ by dual smoothing in the lower dimensional space. Consider an input $\boldsymbol{x} \in \mathbb{R}^d$, which can be partitioned into two spatially non-overlapping sub-inputs, denoted as $\boldsymbol{x}^l \in \mathbb{R}^m$ and $\boldsymbol{x}^l \in \mathbb{R}^n$, where $m+n = d$. Our DRS transforms any base classifier $f$ into a smoothed version $g$ defined as:

$$g(\boldsymbol{x}) = \arg\max_{c \in y} \left( \mathbb{P}(f^l(\boldsymbol{x}^l + \boldsymbol{\varepsilon}^l) = c) + \mathbb{P}(f^r(\boldsymbol{x}^r + \boldsymbol{\varepsilon}^r) = c) \right), \tag{5}$$

where $\boldsymbol{\varepsilon}^l$ and $\boldsymbol{\varepsilon}^r$ are the isotropic Gaussian noises with the mean 0 and the standard deviation $\sigma$ that share the same dimension as $\boldsymbol{x}^l$ and $\boldsymbol{x}^r$. $g(\boldsymbol{x})$ returns whichever the class $f^l$ and $f^r$ are most likely to return by considering the expectation of the probabilities derived from the distributions $\mathcal{N}(\boldsymbol{x}^l + \boldsymbol{\varepsilon}^l; \boldsymbol{x}^l, \sigma^2 \boldsymbol{I})$ and $\mathcal{N}(\boldsymbol{x}^r + \boldsymbol{\varepsilon}^r; \boldsymbol{x}^r, \sigma^2 \boldsymbol{I})$.

**Theorem 2.** *Let $f^l : \mathbb{R}^m \to \mathcal{Y}$ and $f^r : \mathbb{R}^n \to \mathcal{Y}$ be arbitrary deterministic or random functions, and $g$ be the smoothed version defined in Equation 5. Denote $c_A$ and $c_B$ as the most probable and runner-up classes returned by $g$. Then $g(\boldsymbol{x} + \boldsymbol{\delta}) = c_A$ establishes for all adversarial perturbations $\boldsymbol{\delta}$ satisfying that $\|\boldsymbol{\delta}\|_2 \leq R$, where:*

$$R = \frac{\sigma}{\sqrt{2}} \left( \Phi^{-1} \left( p_A^l \right) + \Phi^{-1} \left( p_A^r \right) - 2\Phi^{-1} \left( \frac{\tilde{p}}{2} \right) \right), \text{ where } \tilde{p} = \frac{p_A^l + p_A^r + p_B^l + p_B^r}{2}. \tag{6}$$

In Equation 6, $p^l$ and $p^r$ are the smoothed probabilities returned by base classifier $f^l$ and $f^r$ and $\Phi^{-1}$ is the inverse of the Gaussian CDF. Theorem 2 indicates that the $\ell_2$ certified robustness provided by DRS for the input $\boldsymbol{x}$ is closely linked to the performance of base classifiers $f^l$ and $f^r$ on sub-inputs $\boldsymbol{x}^l$ and $\boldsymbol{x}^r$; the higher the consistency in predictions of $f^l$ and $f^r$ within some specific Gaussian distributions, the greater the level of certified robustness the smoothed classifier $g$ yields.

**Proof of Theorem 2.** In Equation 5, the smoothed classifier $g$ bases its decision on the average of smoothed classification probabilities for $x^l$ and $x^r$. Let $p'$ represent the smoothed probability under the adversarial attack. To mislead the smoothed classifier $g$ successfully, $p'$ must fulfill the condition: $p'^l_B + p'^r_B \geq p'^l_A + p'^r_A$. Let us consider the worst-case scenario for the classifiers under the adversarial attack, where the adversarial attack maximizes the increment in the smoothed probability of the runner-up class, equalizing it with the reduction in the smoothed probability of the most probable class. This leads to the following equations: $p^l_A - p'^l_A = \Delta p^l$, $p^l_B - p'^l_B = -\Delta p^l$, and $p^r_A - p'^r_A = \Delta p^r$, $p^r_B - p'^r_B = -\Delta p^r$, where $\Delta p$ represents the probability change.

Given that $p_B \leq 1 - p_A$ establishes for all scenarios, we can get that:
$$\Phi^{-1}(p_A) - \Phi^{-1}(p_A - \Delta p) \leq \Phi^{-1}(p_B + \Delta p) - \Phi^{-1}(p_B). \tag{7}$$

Thus, according to Theorem 1 and Equation 7, to successfully mislead the smoothed classifier $g$, the adversarial perturbation $\delta^l$ and $\delta^r$ added in $x^l$ and $x^r$ must fulfill that:
$$\left\|\delta^l\right\|_2 + \left\|\delta^r\right\|_2 \geq \sigma \left( \Phi^{-1}\left(p^l_A\right) + \Phi^{-1}\left(p^r_A\right) - \Phi^{-1}\left(p'^l_A\right) - \Phi^{-1}\left(p'^r_A\right) \right). \tag{8}$$

Equation 8 establishes based on using RS to calculate the certified radius in each sub-image. Additional details regarding the derivation of Equation 7 and Equation 8 can be found in Appendix A.2. Assume the critical condition of misleading $g$ is achieved, where $p'^l_B + p'^r_B = p'^l_A + p'^r_A$. We can get:
$$p'^l_A + p'^r_A = \frac{p^l_A + p^r_A + p^l_B + p^r_B}{2} = \tilde{p} \leq 1. \tag{9}$$

Utilizing Equation 9, we can demonstrate that the final two terms in Equation 8 fulfill that:
$$\Phi^{-1}\left(p'^l_A\right) + \Phi^{-1}\left(p'^r_A\right) = \Phi^{-1}\left(p'^l_A\right) + \Phi^{-1}\left(\tilde{p} - p'^l_A\right), \tag{10}$$

and the maximum of Equation 10 is achieved when $p'^l_A = p'^r_A = \frac{\tilde{p}}{2}$ due to $\tilde{p} \leq 1$. So we get:
$$\left\|\delta^l\right\|_2 + \left\|\delta^r\right\|_2 \geq \sigma \left( \Phi^{-1}\left(p^l_A\right) + \Phi^{-1}\left(p^r_A\right) - 2\Phi^{-1}\left(\frac{\tilde{p}}{2}\right) \right) = s, \tag{11}$$

where $s$ is a constant if we get the exact value of the smoothed probability of the most probable and runner-up classes from $f^l$ and $f^r$. Since $x^l$ and $x^r$ are spatially non-overlapping, the overall adversarial perturbation $\delta$ added in the original input $x$ must satisfying the condition:
$$\left\|\delta\right\|_2 = \sqrt{\left\|\delta^l\right\|_2^2 + \left\|\delta^r\right\|_2^2} \in \left[\frac{s}{\sqrt{2}}, s\right]. \tag{12}$$

The proof is concluded if we take the lower bound of Equation 12 as the $\ell_2$ certified robustness provided by DRS. $\square$

Theorem 2 also establishes when give the lower bound estimation $\underline{p^l_A}$ and $\underline{p^r_A}$, along with the upper bound estimation $\overline{p^l_B} = 1 - \underline{p^l_A}$ and $\overline{p^r_B} = 1 - \underline{p^r_A}$. Then the $\ell_2$ certified radius $R$ of DRS is:
$$R = \frac{\sigma}{\sqrt{2}} \left( \Phi^{-1}\left(\underline{p^l_A}\right) + \Phi^{-1}\left(\underline{p^r_A}\right) \right). \tag{13}$$

The following of this paper mainly considers the radius defined in Equation 13 as the $\ell_2$ certified radius provided by DRS for the fair comparison with RS.

### 4.1 ANALYSIS OF THE UPPER BOUND LIMITATION

Assume both RS and DRS take the lower bound estimation for $\underline{p_A}$ and the upper bound estimation for $\overline{p_B}$. We get the relationship of $\ell_2$ certified robustness of input $\overline{x}$ between DRS and RS as follows::
$$R_x = \frac{1}{\sqrt{2}} \left( R'_{x^l} + R'_{x^r} \right). \tag{14}$$

Thus the $\ell_2$ certified radius provided by DRS of an arbitrary $d$-dimensional input $x$ is bounded by:
$$r(x) < \frac{5}{\sqrt{2m}} \Psi^{-1}\left( \frac{\max_{c \in y} P_{\varepsilon^l \sim q^l}(f^l(x^l + \varepsilon^l) = c)}{1 - 5*10^{-7}}; q^l \right) + \frac{5}{\sqrt{2n}} \Psi^{-1}\left( \frac{\max_{c \in y} P_{\varepsilon^r \sim q^r}(f^r(x^r + \varepsilon^r) = c)}{1 - 5*10^{-7}}; q^r \right), \tag{15}$$

where $m + n = d$. This signifies that compared with RS, the DRS, which employs smoothing within the lower-dimensional space, exhibits a significantly more favorable upper bound for $\ell_2$ certified radius, which is proportional to a reduction in the rate by $(1/\sqrt{m} + 1/\sqrt{n})$.

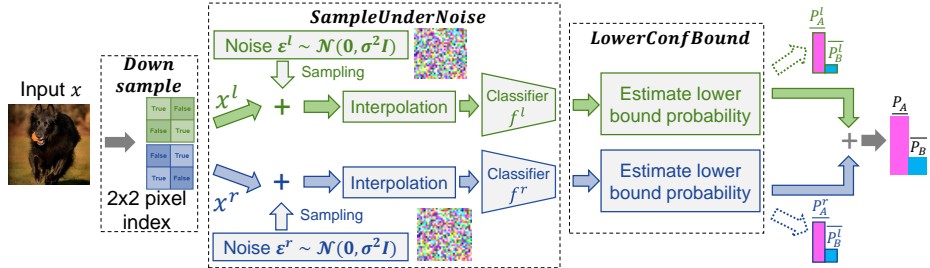

Figure 2: The implementation of Dual Randomized Smoothing (DRS). The image is down-sampled into two non-overlapping sub-images by utilizing two predefined 2x2 pixel indexes.

---

**Algorithm 1** Prediction and $\ell_2$ certified robustness radius of DRS

1: **Input:** base classifiers $\{f^l, f^r\}$, noise standard deviation $\sigma$, image $\boldsymbol{x}$, index $\boldsymbol{idx}$, sampling times $n_0$ and $n$, confidence level $\alpha$
2: $\boldsymbol{x}^l, \boldsymbol{x}^r \leftarrow Downsample(\boldsymbol{x}, \boldsymbol{idx})$
3: $count0^l, count0^r \leftarrow SampleUnderNoise(f^l, f^r, \boldsymbol{x}^l, \boldsymbol{x}^r, n_0, \sigma)$
4: $\hat{c}_A, \hat{c}_B \leftarrow$ top two index in $(count0^l + count0^r)$
5: $count^l, count^r \leftarrow SampleUnderNoise(f^l, f^r, \boldsymbol{x}^l, \boldsymbol{x}^r, n, \sigma)$
6: $\underline{p_A^l}, \underline{p_A^r} = LowerConfBound(count^l[\hat{c}_A], count^r[\hat{c}_A], n, 1 - \alpha)$
7: **if** $\underline{p_A^l} + \underline{p_A^r} \geq 1$ **return** prediction $\hat{c}_A$ and radius $\frac{\sigma}{\sqrt{2}}\left(\Phi^{-1}\left(\underline{p_A^l}\right) + \Phi^{-1}\left(\underline{p_A^r}\right)\right)$
8: **else return** ABSTAIN

---

### 4.2 IMPLEMENTATION OF DUAL RANDOMIZED SMOOTHING USING SPATIAL REDUNDANCY

We implement our dual randomized smoothing as shown in Figure 2. Considering the high information redundancy of adjacent pixels in the image, we utilize two $2 \times 2$ pixel-index matrices as the down-sampling kernels. The two sub-images are generated by sliding the two kernels across the original image, as depicted in Figure 2. Each kernel selectively retains pixels along one of the two diagonals within the $2 \times 2$ image block, moving with a stride of 2. This down-sample approach ensures that the sub-images uphold a significant portion of information from $\boldsymbol{x}$ for classification.

After partition, we first corrupt each sub-image by adding the isotropic Gaussian noise with a zero mean and a standard deviation of $\sigma$. To facilitate fast adaptation for models that are pre-trained on the original images, we employ interpolation to resize the noise-corrupted sub-images to the same spatial resolution as the original image $\boldsymbol{x}$. Then we utilize Algorithm 1 to estimate the lower bound probability and calculate the certified robustness. The function $SampleUnderNoise$ samples a set of $n$ noise-perturbed images and counts the number of samples assigned to each respective class. The function $LowerConfBound$ provides a binomial estimate of the lower bound probability, relying on the number of correctly classified samples $count[\hat{c}_A]$ out of a total of $n$ samples, with confidence of $1 - \alpha$. More details about the sampling and estimation function can be found in Appendix A.6.

## 5 EXPERIMENT RESULT

### 5.1 EXPERIMENT SETUP

**Evaluation details.** Consistent with the prior work, we evaluate the proposed DRS on the CIFAR-10 (Krizhevsky & Hinton, 2009) and ImageNet (Deng et al., 2009) datasets. We report the certified accuracy at each predetermined radius $r$, meaning the percentage of samples that are correctly classified and guaranteed with a certified radius larger than $r$. Moreover, we report the Average Certified Radius (ACR), calculated by averaging the certified radius among all correctly classified samples. We compare the performance of our DRS with RS under different training strategies delicately designed for RS, including Gaussian augmentation (Cohen et al., 2019), consistency regularization (Jeong & Shin, 2020), and diffusion-denoising (Carlini et al., 2023). We also consider the performance of using SmoothAdv (Salman et al., 2019) and Boosting (Horváth et al., 2022) for a more comprehensive comparison.

Table 1: The comparison results of RS and DRS on CIFAR-10. The best performance under each training strategy is bold. We evaluate RS and DRS under two noise levels and report the best result.

| Dataset | Training strategy | Smoothing | $\sigma$ | Certified accuracy at predetermined $\ell_2$ radius $r$ (%) | | | | | | | |
|---|---|---|---|---|---|---|---|---|---|---|---|
| | | | | 0.00 | 0.25 | 0.50 | 0.75 | 1.00 | 1.25 | 1.50 | 2.00 |
| CIFAR-10 | Gaussian | RS | $\{0.25, 0.50\}$ | 76.2 | 60.5 | 42.4 | 33.0 | 22.2 | 14.9 | 9.9 | 0 |
| | | DRS | $\{0.18, 0.36\}$ | **83.4** | **65.8** | **50.2** | 34.5 | **24.7** | 15.8 | 10.5 | 0 |
| | | DRS | $\{0.25, 0.50\}$ | 78.1 | 62.5 | 48.7 | **35.8** | 24.5 | **17.9** | **12.9** | **4.6** |
| | Consistency | RS | $\{0.25, 0.50\}$ | 75.5 | 67.1 | 57.2 | 47.7 | 36.3 | 29.7 | 25.0 | 0 |
| | | DRS | $\{0.18, 0.36\}$ | **77.8** | **70.2** | **59.2** | **47.8** | 37.2 | **31.8** | **26.7** | 0 |
| | | DRS | $\{0.25, 0.50\}$ | 72.5 | 64.7 | 56.6 | 47.1 | **38.6** | 29.2 | 24.1 | **17.4** |
| | Diffusion-denoising | RS | $\{0.25, 0.50\}$ | 83.3 | 72.1 | 57.8 | 44.3 | 31.8 | 24.6 | 18.9 | 0 |
| | | DRS | $\{0.18, 0.36\}$ | **86.7** | **77.6** | 63.1 | 50.6 | **42.4** | 33.7 | 24.8 | 0 |
| | | DRS | $\{0.25, 0.50\}$ | 85.6 | 77.5 | **67.9** | **56.7** | 42.1 | **35.8** | **29.8** | **21.1** |

Table 2: The comparison results of RS and DRS on ImageNet. The best performance under each training strategy is bold. We evaluate RS and DRS under two noise levels and report the best result.

| Dataset | Training strategy | Smoothing | $\sigma$ | Certified accuracy at predetermined $\ell_2$ radius $r$ (%) | | | | | | | |
|---|---|---|---|---|---|---|---|---|---|---|---|
| | | | | 0.00 | 0.25 | 0.50 | 0.75 | 1.00 | 1.25 | 1.50 | 2.00 |
| ImageNet | Gaussian | RS | $\{0.25, 0.50\}$ | 67.0 | 58.6 | 48.4 | 42.6 | 38.0 | 32.4 | 28.4 | 0 |
| | | DRS | $\{0.18, 0.36\}$ | **70.6** | **61.0** | **52.0** | **42.8** | **38.4** | 32.6 | 25.4 | 0 |
| | | DRS | $\{0.25, 0.50\}$ | 67.6 | 58.2 | 49.6 | 42.8 | 35.6 | **33.2** | **29.8** | **21.0** |
| | Consistency | RS | $\{0.25, 0.50\}$ | 64.8 | 58.9 | 54.0 | 48.8 | 42.2 | 36.7 | 35.0 | 0 |
| | | DRS | $\{0.18, 0.36\}$ | **69.2** | **61.6** | **58.4** | **49.0** | **44.1** | **38.4** | 35.2 | 0 |
| | | DRS | $\{0.25, 0.50\}$ | 64.8 | 57.3 | 44.6 | 41.3 | 39.6 | 37.9 | **36.6** | **29.0** |
| | Diffusion-denoising | RS | $\{0.25, 0.50\}$ | 66.8 | 56.4 | 46.2 | 38.0 | 31.2 | 27.6 | 24.4 | 0 |
| | | DRS | $\{0.18, 0.36\}$ | **68.2** | **59.6** | **53.4** | **49.0** | 38.0 | 33.4 | 32.0 | 0 |
| | | DRS | $\{0.25, 0.50\}$ | 65.4 | 58.2 | 52.0 | 45.8 | **41.0** | **33.6** | 28.8 | **24.4** |

**Implementation details.** Following (Cohen et al., 2019; Jeong & Shin, 2020; Horváth et al., 2022), we utilize the ResNet 110 (He et al., 2016) and ResNet 50 as base classifiers for CIFAR-10 and ImageNet under all utilized training strategies. For the diffusion-denoising method (Carlini et al., 2023), we use the same model provided by (Nichol & Dhariwal, 2021): a 50M-parameter model on CIFAR-10 and 552M-parameter class unconditional diffusion model on ImageNet. We re-ensemble the noise-corrupted sub-images for more effective denoising, based on that the adversarial perturbation designed for the classifier has a minor influence on the noise prediction of the diffusion model. For RS, we consider the model trained under Gaussian noise with $\sigma \in \{0.25, 0.50\}$ to get the highest certified accuracy at each predetermined radius $r$. For DRS that gets $\sqrt{2}$ improvement in certified radius when $p_A^l = p_A^r = p_A$, we test its performance under the Gaussian noise with $\sigma \in \{0.18, 0.36\}$ and $\sigma \in \{0.25, 0.50\}$. We report the best performance separately for a more comprehensive and fair comparison. Due to the high computational cost of estimating the smoothed probability, we get our results by evaluating every $5^{th}$ image on CIFAR-10 and every $100^{th}$ image on ImageNet. We set the number of samples $n_0 = 100$, $n = 100,000$, and the estimation confidence parameter $\alpha = 0.001$, meaning that we sample $100,000$ times from the Gaussian distribution to estimate the expectation of the probability and derive the lower bound with the confidence $1 - \alpha$.

**Training details.** For each training strategy, we use the same pre-train model for both low-dimensional classifiers and fine-tune them using the same training strategy as RS. For consistency training, we set the hyperparameter $\lambda$ that controls the weight of consistency loss equal to 10 for both DRS and RS. On CIFAR-10, we fine-tuned all models for a total of 40 epochs using the Adam (Kingma & Ba, 2014) optimizer with an initial learning rate at $1e^{-3}$ and decaying 0.1 at epochs 25 and 35. On ImageNet, we fine-tuned the model for a total of 15 epochs using the Adam optimizer with an initial learning rate at $1e^{-5}$ and decaying 0.1 at epochs 5 and 10.

## 5.2 THE CERTIFIED ACCURACY AT PREDETERMINED RADIUS

We present the comparison results between DRS and RS under the training strategies namely Gaussian augmentation (Cohen et al., 2019), consistency regularization (Jeong & Shin, 2020), and

Table 3: The performance of an identical model using Gaussian augmentation and consistency training strategies on the CIFAR-10 dataset.

(a) Gaussian

| Smoothing | $\sigma$ | Accuracy (%) | ACR |
|---|---|---|---|
| RS | 0.25 | 76.2 | 0.43 |
| DRS | 0.18 | **83.4** | 0.50 |
| DRS | 0.25 | 78.1 | **0.56** |
| RS | 0.50 | 66.4 | 0.54 |
| DRS | 0.36 | **70.8** | **0.58** |
| DRS | 0.50 | 64.4 | 0.60 |

(b) Consistency

| Smoothing | $\sigma$ | Accuracy (%) | ACR |
|---|---|---|---|
| RS | 0.25 | 75.5 | 0.55 |
| DRS | 0.18 | **77.8** | 0.57 |
| DRS | 0.25 | 72.5 | **0.67** |
| RS | 0.50 | 64.3 | 0.74 |
| DRS | 0.36 | **64.8** | 0.77 |
| DRS | 0.50 | 55.3 | **0.81** |

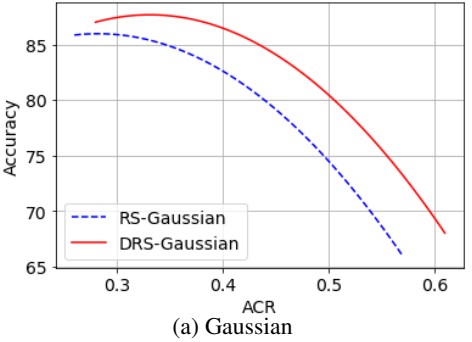

(a) Gaussian

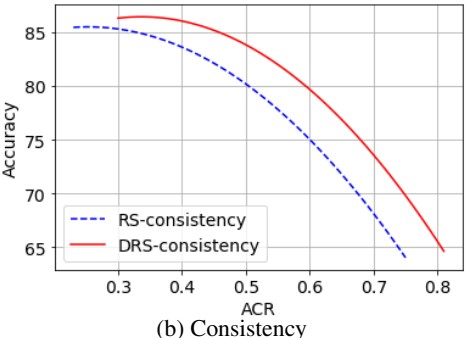

(b) Consistency

Figure 3: The accuracy and robustness trade-off curve of RS and DRS on CIFAR-10 dataset. The data is collected by training multiple models using noise with $\sigma \in [0.07, 0.7]$. We fit this curve by a second-order polynomial function.

diffusion-denoising (Carlini et al., 2023) on the CIFAR-10 and ImageNet in Table 1 and Table 2. The results consistently demonstrate that our DRS significantly outperforms RS in terms of certified accuracy across all radius $r$. Specifically, on the CIFAR-10 dataset, DRS achieves the most substantial gain in certified accuracy at $r = 0.50$, resulting in improvements of approximately 7.8%, 2.0%, and 10.1% when integrated with the three aforementioned methods. On the ImageNet dataset, DRS using the three strategies exhibits the most substantial certified accuracy gain at $r = 0.5$, leading to enhancements of accuracy by approximately 3.6%, 4.4%, and 7.2%.

Furthermore, our investigation reveals that DRS, employing noise levels of $\{0.18, 0.36\}$, attains the best improvement in certified accuracy across all certified radius $r$. This is attributed to that our DRS guarantees a higher bound of the certified radius using the same level of $\sigma$, allowing the classifier to maintain its robustness with a reduced level of noise corruption. While increasing the noise level to $\{0.25, 0.50\}$ causes a slight drop in the certified accuracy due to the inherent trade-off between accuracy and robustness, it guarantees much better certified accuracy for large $\ell_2$ radius, *e.g.*, $r = 2.00$, leading to an overall higher average certified robustness.

## 5.3 ANALYSIS OF THE ACCURACY AND ROBUSTNESS TRADE-OFF BETWEEN RS AND DRS

Whether a fundamental trade-off between accuracy and robustness exists in deep learning models is an active open question. This subsection discusses this trade-off between RS and DRS by reporting the Average Certified Radius (ACR) and the certified accuracy at $r = 0$ as the measure of overall robustness and classification accuracy. We investigate this trade-off in both RS and DRS by examining the performance of an identical model across various noise levels.

We show the classification accuracy and ACR of models trained with Gaussian and consistency regularization on both the CIFAR-10 and ImageNet datasets in Table 3 and Table 4. The results reveal that, in comparison to RS, our DRS consistently increases the classification accuracy significantly while simultaneously improving or preserving the ACR under noise with $\sigma = 0.18 \, or \, 0.36$. When increasing the noise's standard deviation $\sigma$ to 0.25 or 0.50, the DRS significantly improves the ACR but experiences an accuracy drop at $\sigma = 0.50$. We attribute this phenomenon to the high level of noise compromising the utility of the information and subsequently hindering the effective feature

Table 4: The performance of an identical model using Gaussian augmentation and consistency training strategies on the ImageNet dataset.

(a) Gaussian

| Smoothing | $\sigma$ | Accuracy (%) | ACR |
|---|---|---|---|
| RS | 0.25 | 67.0 | 0.48 |
| DRS | 0.18 | **70.6** | 0.51 |
| DRS | 0.25 | 67.6 | **0.62** |
| RS | 0.50 | 57.2 | 0.73 |
| DRS | 0.36 | **61.6** | 0.73 |
| DRS | 0.50 | 54.0 | **0.84** |

(b) Consistency

| Smoothing | $\sigma$ | Accuracy (%) | ACR |
|---|---|---|---|
| RS | 0.25 | 64.8 | 0.52 |
| DRS | 0.18 | **69.2** | 0.53 |
| DRS | 0.25 | 64.8 | **0.64** |
| RS | 0.50 | 56.0 | 0.79 |
| DRS | 0.36 | **59.2** | 0.80 |
| DRS | 0.50 | 54.0 | **0.84** |

Table 5: The performance of boosting DRS by model ensemble on the CIFAR-10 dataset.

| $\sigma$ | Training strategy | model ensemble | Certified accuracy at predetermined $\ell_2$ radius $r$ (%) | | | | ACR |
|---|---|---|---|---|---|---|---|
| | | | 0.00 | 0.25 | 0.50 | 0.75 | |
| 0.18 | Gaussian | 1 | 83.4 | 65.8 | 50.2 | 34.5 | 0.50 |
| | | 2 | **84.7** | **68.8** | **54.6** | **38.6** | **0.53** |
| | consistency | 1 | 77.8 | 70.2 | 59.2 | 47.8 | 0.57 |
| | | 2 | **78.0** | **70.5** | **61.9** | **51.3** | **0.60** |
| | smoothadv | 1 | 75.5 | 69.1 | 56.8 | 45.3 | 0.55 |
| | | 2 | **76.1** | **70.0** | **59.4** | **47.5** | **0.57** |

learning of the base classifier. To give a more straightforward and comprehensive comparison of the accuracy-robustness trade-off inherent in DRS and RS, we trained multiple models on the CIFAR-10 dataset using noise levels with $\sigma \in [0.07, 0.7]$. Figure 3 presents the curve that depicts the trend of accuracy and ACR across models under DRS and RS using various noise levels. Our results illustrate that the proposed DRS consistently enhances certified accuracy and the robustness baseline provided by RS, ultimately achieving a superior accuracy-robustness trade-off.

## 5.4 BOOST DRS BY MODEL ENSEMBLE

Horváth et al. (2022) has demonstrated that the performance of Randomized Smoothing (RS) can be enhanced through model ensemble techniques, which mitigate prediction variance by aggregating decisions from a larger ensemble of models. we apply this boosting approach to DRS by creating ensembles of two models, employing noise with $\sigma = 0.18$ on the CIFAR-10 dataset. Table 5 shows the performance of the boosted DRS using training strategies of Gaussian, consistency, and Smoothadv, indicating that this boosting method can seamlessly integrate with our DRS, and effectively enhance the certified accuracy and average certified robustness.

## 6 CONCLUSION

The curse of dimensionality leads to the diminishing in the $\ell_2$ certified robustness provided by Randomized Smoothing (RS) at a rate of $1/\sqrt{d}$, with $d$ representing the dimension of the input image. To mitigate it, this paper explores the feasibility of providing $\ell_2$ certified robustness for high-dimensional inputs via dual smoothing in the lower-dimensional space. Then a novel smoothing mechanism called Dual Randomized Smoothing (DRS) is proposed, which provides a tight $\ell_2$ certified robustness and yields a superior upper bound for $\ell_2$ robustness. By initially down-sampling the input image into two sub-images, DRS preserves the majority of the input image's information within the low-dimensional data thanks to the information redundancy of the neighboring pixels in most images. Theoretically, we prove a tight $\ell_2$ certified robustness radius for the proposed DRS and demonstrate that DRS achieves a more promising robustness upper bound that decreases at the rate of $(1/\sqrt{m} + 1/\sqrt{n})$, where $m + n = d$. Experimentally, we find that DRS can effectively integrate with existing methods designed for RS and consistently outperforms RS in terms of both the accuracy and $\ell_2$ certified robustness on the CIFAR-10 and ImageNet datasets.

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

# A APPENDIX

## A.1 PROOF OF THEOREM 1

In this subsection, we give the proof of Theorem 1. Assume $f : \mathbb{R}^d \to [0, 1]$ and define the smoothed version $\hat{f}$ as:

$$\hat{f}(\boldsymbol{x}) = \mathop{\mathrm{E}}_{\boldsymbol{\varepsilon} \sim \mathcal{N}(0, I)} [f(\boldsymbol{x} + \boldsymbol{\varepsilon})] = \frac{1}{(2\pi)^{d/2}} \int_{\mathbb{R}^d} f(\boldsymbol{x} + \boldsymbol{\varepsilon}) \exp\left(-\frac{1}{2}\|\boldsymbol{\varepsilon}\|_2^2\right) d\boldsymbol{\varepsilon}. \tag{16}$$

**Lemma 1.** *(Derived from (Salman et al., 2019)), for any function $f : \mathbb{R}^d \to [0, 1]$, subject to the constraint that $\hat{f}(\boldsymbol{x})_c = p$, then $\mu \cdot \nabla \hat{f}(\boldsymbol{x})_c \leq \frac{1}{\sqrt{2\pi}} \exp(-\frac{1}{2}(\Phi^{-1}(p))^2)$ for any unit direction $\mu$.*

Lemma 1 points out that the upper bound of the gradient of the smoothed classifier $\hat{f}$ is limited by $\frac{1}{\sqrt{2\pi}} \exp(-\frac{1}{2}(\Phi^{-1}(p))^2)$. Let $c_A$ and $c_B$ be the most probable and the runner-up classes with probabilities $p_A$ and $p_B$. So, for any $\ell_2$ norm-based adversarial perturbation $\boldsymbol{\delta}$ that successfully mislead the smoothed classifier $\hat{f}$, resulting in $\hat{f}_B(\boldsymbol{x} + \boldsymbol{\delta}) \geq \hat{f}_A(\boldsymbol{x} + \boldsymbol{\delta})$, we can establish:

$$\begin{aligned}
\|\boldsymbol{\delta}\|_2 &\geq \frac{1}{2} \left( \int_{\hat{p}}^{p_A} \left[ \frac{1}{\sqrt{2\pi}} \exp(-\frac{1}{2}(\Phi^{-1}(p))^2) \right]^{-1} dp + \int_{p_B}^{\hat{p}} \left[ \frac{1}{\sqrt{2\pi}} \exp(-\frac{1}{2}(\Phi^{-1}(p))^2) \right]^{-1} dp \right) \\
&= \frac{1}{2} \int_{p_B}^{p_A} \left[ \frac{1}{\sqrt{2\pi}} \exp(-\frac{1}{2}(\Phi^{-1}(p))^2) \right]^{-1} dp \\
&= \frac{1}{2} \Phi^{-1}(p) \Big|_{p_B}^{p_A} \\
&= \frac{1}{2}(\Phi^{-1}(p_A) - \Phi^{-1}(p_B)),
\end{aligned} \tag{17}$$

which concludes the proof of Theorem 1. The detailed proof of lemma 1 can be found in (Salman et al., 2019).

## A.2 PROOF OF THE EQUATION 7 AND EQUATION 8

For Equation 7, as we have $\Phi^{-1}(p) = -\Phi^{-1}(1-p)$, we get:

$$\Phi^{-1}(p_B + \Delta p) - \Phi^{-1}(p_B) = \Phi^{-1}(1 - p_B) - \Phi^{-1}(1 - p_B - \Delta p). \tag{18}$$

Because $1 - p_B \geq p_A$ and we have $\Phi^{-1}(p)$ is a convex function for $p \in [0, 1]$, assume a function $h$ fulfills that:

$$h(p) = \Phi^{-1}(p) - \Phi^{-1}(p - \Delta p), \text{ where } \Delta p \in [0, p], \tag{19}$$

we have the first-order derivative $h'(p) > 0$ establishes for all scenarios. Thus we get:

$$\Phi^{-1}(1 - p_B) - \Phi^{-1}(1 - p_B - \Delta p) \geq \Phi^{-1}(p_A) - \Phi^{-1}(p_A - \Delta p), \tag{20}$$

which concludes the proof of Equation 7. For Equation 8, to make the smoothed probability $p_B$ increases to $p'_B$ after adding adversarial perturbation $\boldsymbol{\delta}$, using Lemma 1, we have:

$$\|\boldsymbol{\delta}\|_2 = \Phi^{-1}(p'_B) - \Phi^{-1}(p_B) \geq \Phi^{-1}(p_A) - \Phi^{-1}(p'_A). \tag{21}$$

Consider the worst case where $p_B = 1 - p_A$, the adversarial perturbations added in each sub-image fulfill that:

$$\begin{aligned}
\|\boldsymbol{\delta}^l\|_2 + \|\boldsymbol{\delta}^r\|_2 &= \Phi^{-1}(p'^l_B) + \Phi^{-1}(p'^r_B) - \Phi^{-1}(p^l_B) - \Phi^{-1}(p^r_B) \\
&= \Phi^{-1}(p^l_A) + \Phi^{-1}(p^r_A) - \Phi^{-1}(p'^l_A) - \Phi^{-1}(p'^r_A)
\end{aligned} \tag{22}$$

which conclude the proof of Equation 8.

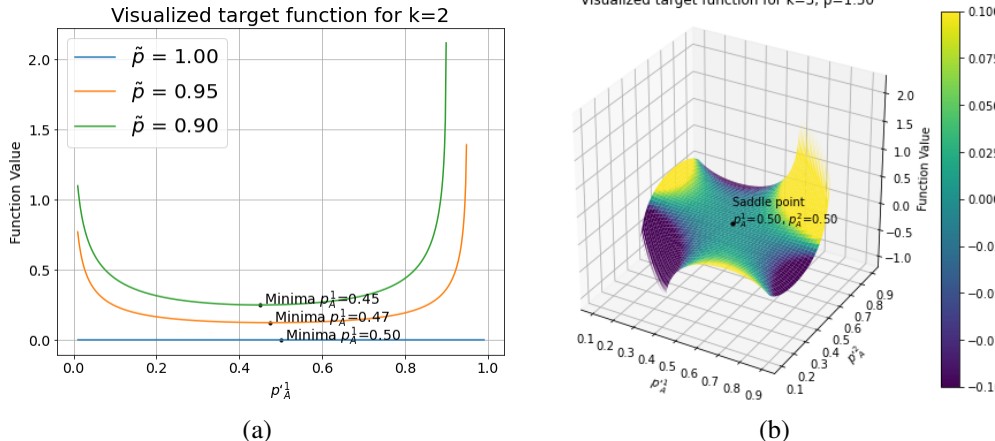

Figure 4: (a) The visualized landscape of objective function for $k = 2$ across various $\tilde{p}$. (b)The visualized landscape of objective function for $k = 3$ with $\tilde{p} = 1.50$.

### A.3 ANALYSIS OF K-PARTITIONING BASED SMOOTHING

Section 4.1 illustrates that dual smoothing in the lower-dimensional space effectively mitigates the curse of dimensionality. This section explores the feasibility of further enlarging the certified robustness upper bound by k-partitioning based smoothing.

Consider an input $x \in \mathbb{R}^d$, which can be partitioned into $k$ spatially non-overlapping sub-inputs, denoted as $x^j \in \mathbb{R}^{d_j}$, where $\sum_{j=1}^{k} d_j = d$. The k-partitioning based smoothing transforms any classifier $f$ into a smoothed version $g$ defined as:

$$g(x) = \underset{c \in y}{\arg\max} \sum_{j=1}^{k} \mathbb{P}(f^j(x^j + \varepsilon^j) = c), \tag{23}$$

where $\varepsilon^j$ is the isotropic Gaussian noises with the mean 0 and the standard deviation $\sigma$ that share the same dimension as $x^j$. Denote $c_A$ and $c_B$ as the most probable and runner-up classes returned by $g$. Assuming $p^j$ is the smoothed probability returned by the base classifier $f^j$ and $\delta^j$ is the adversarial perturbation adding in sub-input $x^j$. To mislead the smoothed classifier $g$ successfully, according to Equation 17, the adversarial perturbation $\delta^j$ added in each sub-input $x^j$ must fulfill:

$$\left\| \delta^j \right\|_2 \geq \sigma \left( \Phi^{-1}\left( p^j_A \right) - \Phi^{-1}\left( p'^j_A \right) \right), \quad \forall j \in \{1, \dots, k, \}. \tag{24}$$

Thus the sum of the $\delta^j$ fulfills:

$$\sum_{j=1}^{k} \left\| \delta^j \right\|_2 \geq \sigma \sum_{j=1}^{k} \left( \Phi^{-1}\left( p^j_A \right) - \Phi^{-1}\left( p'^j_A \right) \right), \tag{25}$$

where $p'^j_A$ represents the smoothed probability under the adversarial attack and satisfies that $p^j_A \geq p'^j_A$ and $\sum_{j=1}^{k} p'^j_A = \frac{1}{2} \sum_{j=1}^{k} \left( p^j_A + p^j_B \right)$. In Equation 25, $p^j_A$ is a constant value that can be estimated by $n$ sampling. Thus calculating the certified robustness of smoothed classifier $g$ can be turned into solving the optimization problem defined as:

$$\begin{aligned}
\min_{p'^j_A} \quad & -\sum_{j=1}^{k} \Phi^{-1}\left( p'^j_A \right) \\
\text{s.t.} \quad & \sum_{j=1}^{k} p'^j_A = \frac{1}{2} \sum_{j=1}^{k} (p^j_A + p^j_B), \\
& p'^j_A \leq p^j_A, \quad \forall j \in \{1, \dots, k\}.
\end{aligned} \tag{26}$$

However, due to $\Phi^{-1}$ being non-convex, the above problem is not a typical convex optimization.

- For $k = 2$, denote:

$$\tilde{p} = \frac{1}{2}(p_A^1 + p_B^1 + p_A^2 + p_B^2) \leq 1, \tag{27}$$

where $\tilde{p}$ represents the sum of the average probability between the most probable class $C_A$ and the runner-up class $C_B$. According to the first constraint in Equation 26, we can derive that $p'^2_A = \tilde{p} - p'^1_A$. Meanwhile, $\Phi^{-1}\left(p'^2_A\right) = -\Phi^{-1}\left(1 - p'^2_A\right)$ establishes for $p'^2_A \in [0, 1]$. Thus, the above objective function can be transformed into:

$$\min_{p'^j_A} \quad \Phi^{-1}\left(p'^1_A + 1 - \tilde{p}\right) - \Phi^{-1}\left(p'^1_A\right). \tag{28}$$

This The objective function Equation 28 is a convex function with a global minimum achieved by $p'^1_A = p'^2_A = \frac{1}{2}\tilde{p}$.

- For $k > 2$, denote:

$$\tilde{p} = \frac{1}{2}\sum_{j=1}^{k}(p_A^j + p_B^j) \leq \frac{k}{2}. \tag{29}$$

The objective function is non-convex. Providing a numerically stable solution for the global minimum under these circumstances is challenging (the minimum might not exist). A saddle point is achieved by:

$$p'^j_A = \tilde{p}/k, \forall j \in \{1, \ldots, k\}. \tag{30}$$

The landscape of objective function for $k = 2$ and $k = 3$ is visualized in Figure 4 for a more straightforward comparison and illustration.

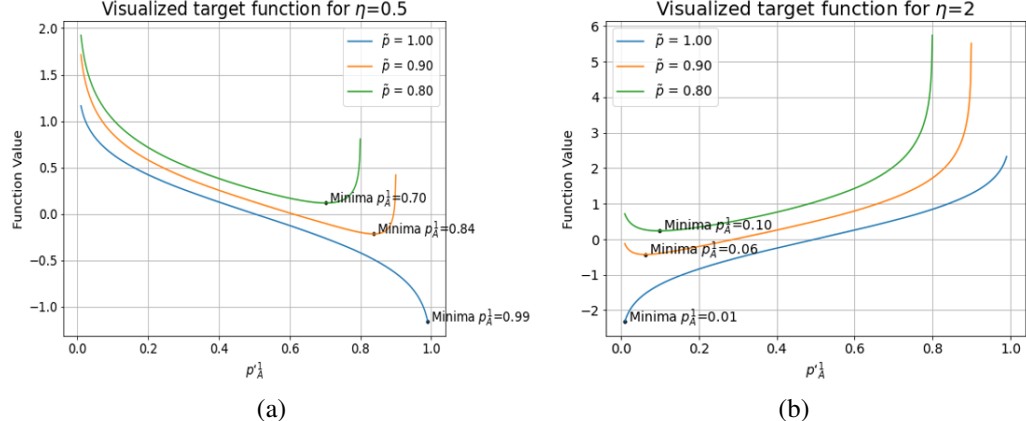

Figure 5: The visualized landscape of objective function and for smoothing with different variances. For symmetric cases where $\eta = 2$ and $\eta = 0.5$, the sum of the two optimal $p'^l_A$ is $\tilde{p}$.

### A.4 ANALYSIS OF SMOOTHING WITH DIFFERENT VARIANCE

This subsection discusses the feasibility of using Gaussian noise with different variances to smooth the sub-images in DRS. Consider an input $x \in \mathbb{R}^d$, which can be partitioned into two spatially non-overlapping sub-inputs, denoted as $x^l \in \mathbb{R}^m$ and $x^r \in \mathbb{R}^m$. Define the smoothed classifier $g$ same as in Equation 5, while $\varepsilon^l$ and $\varepsilon^r$ are the isotropic Gaussian noises with the mean 0 and the standard deviation $\sigma^l$ and $\sigma^r$. According to Equation 24 and 25, to mislead $g$ successfully, the adversarial perturbation $\delta^j$ must fulfill:

$$\left\| \delta^l \right\|_2 + \left\| \delta^r \right\|_2 \geq \sigma^l \left( \Phi^{-1} \left( p^l_A \right) - \Phi^{-1} \left( p'^l_A \right) \right) + \sigma^r \left( \Phi^{-1} \left( p^r_A \right) - \Phi^{-1} \left( p'^r_A \right) \right), \tag{31}$$

where $p'_A$ represents the smoothed probability under the adversarial attack and satisfies that $p_A \geq p'_A$ and $p'^l_A + p'^r_A = \frac{1}{2} \left( p^l_A + p^l_B + p^r_A + p^r_B \right)$. Assume $\eta = \sigma^r / \sigma^l$, Equation 31 can be rewritten as:

$$\left\| \delta^l \right\|_2 + \left\| \delta^r \right\|_2 \geq \sigma^l \left( \Phi^{-1} \left( p^l_A \right) + \eta \Phi^{-1} \left( p^r_A \right) \right) - \sigma^l \left( \Phi^{-1} \left( p'^l_A \right) + \eta \Phi^{-1} \left( p'^r_A \right) \right). \tag{32}$$

Thus according to Equation 26 and Equation 28, calculating the certified robustness of smoothed classifier $g$ can be turned into solving the optimization problem defined as:

$$\min_{p'^l_A} \quad \eta \Phi^{-1} \left( p'^l_A + 1 - \tilde{p} \right) - \Phi^{-1} \left( p'^l_A \right), \tag{33}$$
$$\text{s.t.} \quad \tilde{p} \in (0, 1), \quad \eta > 0.$$

Denote $\tilde{p} = \frac{1}{2} (p^1_A + p^1_B + p^2_A + p^2_B)$. The first-order derivative of the above objective function is:

$$\eta \frac{d}{d\, p'^l_A} \Phi^{-1} \left( p'^l_A + 1 - \tilde{p} \right) - \frac{d}{d\, p'^l_A} \Phi^{-1} \left( p'^l_A \right)$$
$$\Rightarrow \sqrt{2\pi} \left( \eta \exp \left( \frac{\Phi^{-1} \left( p'^l_A + 1 - \tilde{p} \right)^2}{2} \right) - \exp \left( \frac{\Phi^{-1} \left( p'^l_A \right)^2}{2} \right) \right). \tag{34}$$
$$\Rightarrow \sqrt{2\pi} \eta \exp \left( \frac{\Phi^{-1} \left( p'^l_A \right)^2}{2} \right) \left( \exp \left( \frac{\Phi^{-1} \left( p'^l_A + 1 - \tilde{p} \right)^2 - \Phi^{-1} \left( p'^l_A \right)^2}{2} \right) - \frac{1}{\eta} \right)$$

Owing to the inherent positivity of the exponential function, the sign of the first derivative hinges on the comparison of the value of $\Phi^{-1} \left( p'^l_A + 1 - \tilde{p} \right)^2 - \Phi^{-1} \left( p'^l_A \right)^2$ with the critical value of

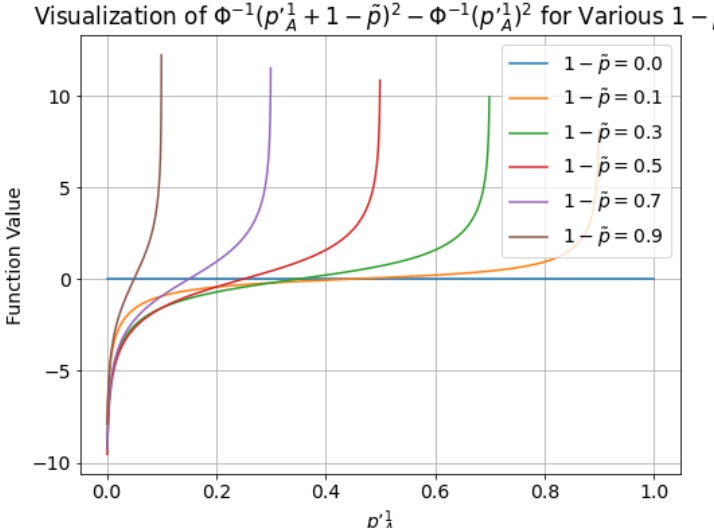

Figure 6: The visualized landscape of $\Phi^{-1}\left(p'^l_A + 1 - \tilde{p}\right)^2 - \Phi^{-1}\left(p'^l_A\right)^2$.

$2 \ln \frac{1}{\eta}$. Moreover, the expression $\Phi^{-1}\left(p'^l_A + 1 - \tilde{p}\right)^2 - \Phi^{-1}\left(p'^l_A\right)^2$ exhibits a monotonic increasing trend from negative infinity to positive infinity within the interval $\tilde{p} \in (0, 1)$ (a horizontal line if $\tilde{p} = 1$.). The visualization of this function is shown in Figure 6 for illustration. Therefore, it can be asserted that the objective function initially demonstrates a monotonically decreasing behavior, which subsequently transitions into an increasing trend. Thus, the minimum is achieved when the first-order derivative equals zero, which is:

$$\Phi^{-1}\left(p'^l_A + 1 - \tilde{p}\right)^2 - \Phi^{-1}\left(p'^l_A\right)^2 = 2 \ln \frac{1}{\eta}. \tag{35}$$

When $\tilde{p}$ is set to 1, the objective function assumes a monotonic property (specifically, a decreasing trend for $\eta > 1$ and conversely, an increasing trend for $\eta < 1$), with the minimum being attained at the boundary of the domain of $p'^j_A$. Thus according to Equation 11 and 12, we can derive a certified robustness boundary for DRS under different variances smoothing. The landscape of the objective function is visualized in Figure 5 for a more straightforward illustration.

## A.5 JUSTIFICATION OF THE TIGHTNESS OF DRS ROBUSTNESS BOUND

The tightness of the certified robustness of RS is proven by Cohen et al. (2019). Assume the certified radius directly deduced from RS is tight. Given that $p_B = 1 - p_A$, the inequalities presented in Equations 7, 8, 9, and 11 are transformed into equalities. This indicates the smallest of $\left\|\delta^l\right\|_2 + \left\|\delta^r\right\|_2$ to break DRS is $\sigma\left(\Phi^{-1}\left(p^l_A\right) + \Phi^{-1}\left(p^r_A\right)\right)$. Denote $s = \left\|\delta^l\right\|_2 + \left\|\delta^r\right\|_2$, according to Equation 12, the adversarial perturbation $\delta$ is:

$$\|\boldsymbol{\delta}\|_2 = \sqrt{\left\|\boldsymbol{\delta}^l\right\|_2^2 + \left\|\boldsymbol{\delta}^r\right\|_2^2} \in \left[\frac{s}{\sqrt{2}}, s\right]. \tag{36}$$

Consequently, in the worst case situation that $\left\|\delta^l\right\|_2 = \left\|\delta^r\right\|_2$, the largest certified robutsness provided by DRS is $R = \frac{\sigma}{\sqrt{2}}\left(\Phi^{-1}\left(p^l_A\right) + \Phi^{-1}\left(p^r_A\right)\right)$, which justify the tightness of DRS robustness bound.

## A.6 DETAILS ABOUT THE DRS CERTIFIED ALGORITHM

This section illustrates the details of the certified algorithm of DRS.

$Downsample(\boldsymbol{x}, \boldsymbol{idx})$ down-samples the original image $\boldsymbol{x}$ into two spatially non-overlapping sub-images based on the index vector $\boldsymbol{idx}$ and returns the two sub-images at dimension $m$ and $n$.

$SampleUnderNoise(f^l, f^r, \boldsymbol{x}^l, \boldsymbol{x}^r, n)$ initially performs $n$ times sampling from the distribution of $\mathcal{N}(\boldsymbol{x}^l + \boldsymbol{\varepsilon}^l; \boldsymbol{x}^l, \sigma^2 \boldsymbol{I})$ and $\mathcal{N}(\boldsymbol{x}^r + \boldsymbol{\varepsilon}^r; \boldsymbol{x}^r, \sigma^2 \boldsymbol{I})$. Then it counts the frequency of predicted classes by $f^l$ and $f^r$ on those $n$ samples and returns two corresponding $k$ dimensional arrays that contain these frequency counts.

$LowerConfBound(count^l[\hat{c}_A], count^r[\hat{c}_A], n, 1 - \alpha)$ calculates the lower bound probability $p$ with confidence at least $1 - \alpha$ using binomial estimation. Where $p$ is an unknown probability fulfilling that $count(\hat{c_A}) \sim \mathcal{B}(n, p)$, where is the $\mathcal{B}$ binomial distribution with probability $p$ and sampling times $n$.

