# OpenReview forum: "Mitigating the Curse of Dimensionality for Certified Robustness via Dual Randomized Smoothing"
_ICLR.cc/2024/Conference — ICLR 2024 poster_

### Official Review · Reviewer_Vvvf · 2023-10-24

**Soundness:** 3 good
**Presentation:** 4 excellent
**Contribution:** 2 fair
**Rating:** 6
**Confidence:** 3

**Summary:**

The authors introduced the dual random smoothing technique in order to mitigate the curse of dimensionality in the certified robustness bound in the realm of RS techniques.

**Strengths:**

1. By partitioning the input into two orthogonal spaces, and apply RS to each of them, the authors mitigated the curse of dimensionality.
2. The technique proposed can be used in conjunction with a number of other methods, outside of plain RS.
3. Theorem 2 showcases that this kind of technique is agnostic of the exact partition scheme, which make the application of this technique more ubiquitous.

**Weaknesses:**

1. This paper does not convince me that DRS will be better at RS in in terms of classification accuracy. Although experiments can partially validate this idea, but I think some form of theoretical analysis is needed in order to explain that DRS is at least not a lot worse than RS, because experiments can be easily tuned to show better results.

2. The authors did not explain the partitioning mechanism in too much detail, so I'm curious whether a different partitioning mechanism will yield drastically different classification results (in terms of dimensionality scaling it should be fine due to Theorem 2), because this is very possible. Also the authors should explain how to choose a good partitioning scheme as this is conceivably very important to DRS.

**Questions:**

If partitioning into 2 spaces is good for mitigating curse of dimensionality, what about 3 spaces and even more? theoretically speaking they should mitigate the problem better, but why choose 2?

---

> ### Author Response · Authors · 2023-11-17
> **Resopnse to Reviewer Vvvf**
>
> We thank the reviewer for taking the time to read this paper and giving constructive feedback. Please find our response below.
>
> > #### **W1: The classification accuracy of DRS and RS. Some form of theoretical analysis is needed in order to explain that DRS is at least not a lot worse than RS.**
>
> The classification accuracy of DRS compared to that of RS is data-dependent. In the case of high-dimensional, real-world datasets, particularly those with significant information redundancy like images, the classification accuracy of DRS can be very close to that of RS. This comes from a well-known fact that images only occupy a very small subspace of the image space spanned by pixels, especially for high-resolution images. Hence, each sub-sample obtained through down-sampling retains a substantial portion of the original information. [1] claims that masking a high proportion of the input image, e.g., 75%, yields a meaningful self-supervisory task. In general, the classification accuracy of DRS will not be a lot worse than RS for those data with high information redundancy if the data is properly partitioned.
>
> To verify that, we conducted a comparative experiment to evaluate the classification accuracy on the CIFAR-10 dataset using $f$, $f^l$,$f^r$, and $f^l+f^r$. For all classifiers, we employed a standard training strategy; we used the ResNet110 as the classification model, and SGD as the optimizer with  $learningrate =0.1,~ momentum=0.9,~ and~weightdecay=5*10^{-4}$. We used the standard augmentation including random crop and flip. Each model was trained for 100 epochs with a learning rate decay implemented via $CosineAnnealingLR$. The experiment result is shown in the following table. The result reveals that DRS shows a very similar accuracy performance to RS for this image dataset.
>
> | Model     | Classification Accuracy (%)             |
> | --------- | -------------------- |
> | $f$      | $94.0~(\pm0.1)$ |
> | $f^l+f^r$ | $93.9~(\pm0.2)$ |
> | $f^r$    | $92.8~(\pm0.1)$ |
> | $f^l$     | $92.8~(\pm0.2)$ |
>
> For the improvement in the certified robustness of DRS, we primarily attribute it to its ability to guarantee the same robustness as RS using a lower noise variance. In Tables 3 and 4, we show that under consistency training, DRS under a lower noise variance achieves a higher certified accuracy and upholds a comparable level of certified robustness as RS, but encounters some accuracy drop when the noise variance increases to match that of RS.
>
> > #### **W2: Whether a different partitioning mechanism will yield a drastically different classification result. Explain how to choose a good partitioning scheme.**
>
> Yes, a different partitioning mechanism may yield a drastically different classification result. A good partition scheme depends on the structure of the data sample. In general, for most real-world datasets, down-sampling is a good partitioning scheme because it is a good way to let each of the partitioned data contain information close to that in the original data, thereby preserving classification accuracy.
>
>
> > #### **Q1: What about 3 spaces and even more partitioning.**
>
> Further partitioning the input is quite promising for enhancing the robustness upper bound. Reflecting on your discerning insight, in Appendix A.3 of the revised manuscript, we further explore the feasibility of k-partitioning based smoothing. By deriving a general architecture of the smoothed classifier $g$ using $k$-partitioning, in Equation (26), we reformulate the guarantee of certified robustness as an optimization problem. Our findings in Appendix A.3 indicate that:
>
> 1. for $k=2$,  this optimization is proven to be convex, where the global minimum to guarantee the certified robustness is achieved by $\mathop{p'}\nolimits_A^1=\mathop{p'}\nolimits_A^2=\frac{1}{2}\tilde p$. This is in alignment with our assertion in Equation (11).
> 2. For $k>2$, this optimization is non-convex. Providing a numerically stable solution for the global minimum under these circumstances is challenging (the minimum might not exist). A saddle point is achieved when $\mathop{p'}\nolimits_A^j  = {{\tilde p} \mathord{\left/{\vphantom {{\tilde p} k}} \right.} k} , \forall j \in \{1, \ldots, k\}$, but it does not strictly ensure the certified robustness.
>
> We present an in-depth analysis and an illustrative visualization of this $k$-partitioning based smoothing in Appendix A.3, which may help future research endeavors.
>
> > [1] Masked autoencoders are scalable vision learners. He, Kaiming, et al., CVPR, 2022.

---

> ### Comment · Reviewer_Vvvf · 2023-11-22
>
> I want to thank the authors for their detailed response. I think I'm still not entirely sold on the idea that DRS does not sacrifice classification accuracy too much, and the explanation is a bit hand wavy. However, I do appreciate the author's additional experiments and clear illustration. I think given the current experiments and explanation I would like to raise my score to 6, since I do see value in DRS. However as a follow-up work or a journal version it is highly desirable to have some theoretical guarantee surrounding the accuracy of DRS. As for partitioning scheme, I understand down-sampling is nice, but what kind of down-sampling is critical (different modes of anti-aliasing and all that). However these details can be left to future works.

---

> > ### Author Response · Authors · 2023-11-22
> >
> > Dear Reviewer Vvvf,
> >
> > We sincerely appreciate your constructive and positive feedback. Your insights and suggestions are invaluable and will greatly assist us in enhancing our future work.
> >
> > Best Regards,
> > Authors of Paper 1829

---

### Official Review · Reviewer_iAVM · 2023-10-30

**Soundness:** 3 good
**Presentation:** 3 good
**Contribution:** 3 good
**Rating:** 8
**Confidence:** 3

**Summary:**

This paper proposes "dual randomized smoothing" as a method to alleviate the curse of dimensionality faced by standard randomized smoothing. The proposed method splits inputs into two lower-dimensional components, then performs smoothing in the low-dimensional spaces before bringing the components together to form a final prediction. A certified radius of robustness is theoretically proven, and experiments on benchmark image datasets are given to showcase the enhanced robustness of the method.

**Strengths:**

1. The introduction is thorough yet concise, and very nicely recapitulates the curse of dimensionality for randomized smoothing and clearly states the paper's contributions towards alleviating these issues.
2. The related works section is thorough and well situates the proposed method within the vast amounts of works focused on enhancing randomized smoothing.
3. The main theoretical result (robust radius for the DRS method) appears sound and provides novel insight into the dimensionality aspects at hand.
4. The experiments are extensive and show nice improvements over standard RS.
5. Overall, the paper is well-written and is very easy to read.

**Weaknesses:**

1. The authors claim that prior works fail to mitigate the curse of dimensionality for randomized smoothing approaches. However, they have missed the relevant works [1],[2], which should be discussed.
2. The authors propose to use interpolation in Section 4.2 to utilize pre-trained models defined on the original input space, to act as the lower-dimensional classifiers. It is not clear whether they are using the same pre-trained model for both low-dimensional classifiers, or two different ones. Further, the potential downsides to using such interpolation schemes is not discussed. How much performance loss is there in taking this approach when compared to using two models $f^l$ and $f^r$ specifically trained for this dual RS approach? Of course, this would require additional pre-training, which may be costly for certain applications, but it would be good to know how much better the corresponding robustness certificates would be.
3. First word in Section 5.3 is missing capitalized first letter.
4. I think a bit more explanation would be useful on how the down-sampling is performed to break each input into its two parts. Specifically, it is sort of glossed over in text and left to the reader to interpret the procedure based on Figure 2. A simple one to two sentence explanation in the main body of the paper would suffice, preferably with an explicit pointer to the graphic of Figure 2, where the authors may want to state that the index set corresponds to a two-part partition of the images pixel indices, and $x^l$ is the vector composed of $x$'s pixel values $x_i$ with $i\in\text{idx}$, and $x^r$ is the vector composed of $x$'s pixel values $x_i$ with $i\notin\text{idx}$.

[1] Projected Randomized Smoothing for Certified Adversarial Robustness, Pfrommer et al., TMLR, 2023
[2] Intriguing Properties of Input-Dependent Randomized Smoothing, Sukenik et al., ICML, 2022

**Questions:**

1. Where does the $5*10^{-7}$ come from in (4)? Is this from a pre-specified sample size in RS? It would be nice to briefly mention this in Section 3.2.
2. Why restrict the two low-dimensional smoothing procedures to share the same variance? How restrictive/suboptimal is this in practice? Will your guarantees generalize to the setting with different smoothing variances?
3. The authors claim that their bound is tight. Perhaps I missed this, but where is this claim proven/justified?

---

> ### Author Response · Authors · 2023-11-17
> **Response to Reviewer iAVM--Part 1**
>
> Thank you so much for taking the time to read this paper and giving constructive feedback. Please find our response below.
>
> > #### **W1: The authors claim that prior works fail to mitigate the curse of dimensionality for randomized smoothing approaches. However, they have missed the relevant works [1],[2], which should be discussed.**
>
> Following your suggestion, we modified our claim in our revised version as "Prior works have not adequately tackled the challenge of the curse of dimensionality in the context of high-dimensional inputs.", and added the discussion of these two relevant works in our related work as "Sukenik et al. (2022) prove that the input-dependent RS also suffers from the curse of dimensionality and Pfrommer et al. (2023) propose a projected randomized smoothing that enhances the lower bound on the certified volume.".
>
> Projected Randomized Smoothing (PRS) innovatively linearly projects inputs into a lower-dimensional space and improves the lower bound on the certified volume. However, PRS could be vulnerable to some specific perturbation directions. In contrast, DRS maintains efficacy across all perturbation directions. Additionally, DRS maintains its effectiveness for higher-dimensional datasets, that are profoundly impacted by the curse of dimensionality, while PRS might fail due to the large memory for holding the entire dataset. Sukenik et al. prove that input-dependent randomized smoothing suffers from the curse of dimensionality, indicating its effectiveness primarily for small to medium-sized inputs. However, this paper does not propose the solution for this curse.
>
> > #### **W2: It is not clear whether they are using the same pre-trained model for both low-dimensional classifiers. The performance of using a model specifically trained for this DRS approach? Discuss the potential downsides of using interpolation schemes.**
>
> 1. For each training strategy discussed in our paper (such as Gaussian augmentation, consistency $\ldots$), we use the same pre-trained model for both low-dimensional classifiers and fine-tune them using the same training strategy as RS for a fair comparison. We added this clarification in our revised manuscript.
> 2. An intriguing observation is a parallelism in performance trends between DRS and RS across diverse training strategies and models. For example, in DRS, using a stronger pre-trained model based on consistency regularization outperforms using the pre-trained model based on Gaussian augmentation. Thus we think enhancing DRS with a stronger specifically trained model is promising.
> 3. Employing interpolation slightly elevates the computational complexity, but it helps the alignment of the training data distributions in DRS and RS, making DRS well adapt to existing training strategies designed for RS.
>
> > #### **W3 and 4: The first word in section 5.3 is not capitalized and more explanation on how the down-sampling is performed.**
>
> In our revised version, we corrected this typo and added an explanation of the down-sampling process: "Considering the high information redundancy of adjacent pixels in the image, we utilize two $2\times2$ pixel-index matrices as the down-sampling kernels. Each kernel selectively retains pixels along one of the two diagonals within the $2\times2$ image block, as depicted in Figure 2. The two sub-images are generated by sliding the two kernels across the original image, moving with a stride of 2.".
>
> > [1] Projected Randomized Smoothing for Certified Adversarial Robustness, Pfrommer et al., TMLR, 2023.
>
> > [2] Intriguing Properties of Input-Dependent Randomized Smoothing, Sukenik et al., ICML, 2022.
>
> > [3] Certified adversarial robustness via randomized smoothing, Cohen et al., ICML, 2019.

---

> ### Author Response · Authors · 2023-11-17
> **Response to Reviewer iAVM--Part 2**
>
> > #### Q1: Explain the $5*{10^{ - 7}}$ in Equation (4). Is this from a pre-specified sample size in RS?
>
> In our revised manuscript, we added an explanation that "the parameter $5 \times 10^{-7}$ is derived from a predefined error margin ($1 \times 10^{-6}$) between the estimated probability and true probability over the origin-symmetric distribution.". Indeed, this parameter maintains a relationship with the sample size.
>
> > #### **Q2: Will DRS guarantee to generalize to the setting with different smoothing variances? Does restricting the same variance make DRS suboptimal in practice?**
>
> 1. Yes, DRS guarantees generalization to the setting with difference smoothing variance. Following your insights, we analyze the generalizability of DRS with different smoothing variances in Appendix A.4 of the revised manuscript. Let $\eta$ denote the ratio of standard deviations selected for two sub-inputs, specifically $\eta={{\sigma} ^r}/{{\sigma} ^l}$. Our finding in Appendix A.4 is that the calculation of the worst-case robustness provided by DRS with different smoothing variances can be formulated into a convex optimization problem. A global minimum that guarantees the certified robustness of DRS is achieved by making the first-order derivative equal to zero, which is:
>    $$\eta *\frac{d}{{{d{\mathop{p'}\nolimits_A^l}}}}{\Phi ^{ - 1}}\left( {\mathop{p'}\nolimits_A^l + 1 - \tilde p} \right) = \frac{d}{{{d{\mathop{p'}\nolimits_A^l}}}}{\Phi ^{ - 1}}\left( \mathop{p'}\nolimits_A^l \right).$$
>    For $\eta=1$, the solution of the above equation is $\mathop{p'}\nolimits_A^l=\mathop{p'}\nolimits_A^r=\frac{1}{2}\tilde p$, which is in alignment with our assertion in Equation (11). An illustrative visualization of this problem is presented in Figure 5. Please kindly refer to the updated manuscript for more details.
> 2. Yes, the reviewer is correct that restricting the uniform variance for the two sub-images is suboptimal in practice. Nonetheless,  the substantial information overlap in data conveyed by the sub-images indicates that their respective optimal variance levels are close. Inspired by the input-dependent RS [2], determining the optimal variance for each image within the DRS framework is promising work.
>
> > #### **Q3: Explain the tightness of DRS robustness bound**
>
> The tightness of the certified robustness of RS is proven in [3]. For DRS, we assume that the certified radius directly derived from RS is tight. Given that ${p_B} =  1 - {p_A}$,  the inequalities presented in Equations (7), (8), (9), and (11) are transformed into equalities. This indicates the smallest of ${\left\| {{{\delta} ^l}} \right\|_2} + {\left\| {{{\delta} ^r}} \right\|_2}$ to break DRS is $\sigma \left( {{\Phi ^{ - 1}}\left( {p_A^l} \right) + {\Phi ^{ - 1}}\left( {p_A^r} \right)}\right)$. Consequently, in the worst case situation that ${\left\| {{{\delta} ^l}} \right\|_2} = {\left\| {{{\delta} ^r}} \right\|_2}$, the largest certified robustness provided by DRS is $R=\frac{\sigma }{{\sqrt 2 }}\left( {{\Phi ^{ - 1}}\left( { {p_A^l} } \right) + {\Phi ^{ - 1}}\left( {{p_A^r} } \right)} \right)$, which proves the tightness of DRS robustness bound. We added this justification in our revised manuscript in Appendix 5.

---

> > ### Comment · Reviewer_iAVM · 2023-11-17
> >
> > Thank you to the authors for their very thorough responses and for their updates to the manuscript. The additions of Appendices A.3--A.5 are particularly nice.
> >
> > One final comment I have is that you state that "the worst-case robustness provided by DRS with different smoothing variances can be formulated into a convex optimization problem", and refer to Figure 5. However, based on Figure 5, it appears that sometimes the optimization is nonconvex (the blue colored objective function plots). Maybe I am misinterpreting something, but if it is the case that the optimization is sometimes nonconvex, then you should ensure your language reflects this, and clarify in the manuscript whether $\tilde{p}=1$ is the only case of nonconvexity or not.
> >
> > Overall, I appreciate the improvements, and will therefore increase my score by one step in the ICLR scoring rubric (from 6 to 8, since 7 is not an option).

---

> > > ### Author Response · Authors · 2023-11-19
> > > **Response to Reviewer iAVM**
> > >
> > > Dear Reviewer iAVM,
> > >
> > > Thank you so much for your positive and constructive feedback, which is very helpful and makes our paper stronger! We are glad that our responses address your concern. We have modified our claim in the newly updated manuscript and given a more detailed analysis to prove our assertion that 'When $\tilde p \in (0,1)$, the minimum is achieved when the first-order derivative equals zero' and "When $\tilde p$ is set to 1, the objective function assumes a monotonic property (specifically, a decreasing trend for $\eta > 1$ and conversely, an increasing trend for $\eta < 1$), with the minimum being attained at the boundary of the domain of $\mathop{p'}\nolimits_A^j$.". Please kindly refer to Equation (34),  Equation (35) and Figure 6 in Appendix A.4 for more details (marked in lightblue). We are always available and eager to address any additional questions you might have during our discussion.
> > >
> > > Best regards,
> > >
> > > Authors of Paper1829

---

### Official Review · Reviewer_h68g · 2023-10-31

**Soundness:** 3 good
**Presentation:** 3 good
**Contribution:** 3 good
**Rating:** 8
**Confidence:** 1

**Summary:**

-

**Strengths:**

-

**Weaknesses:**

-

**Questions:**

-

---

> ### Author Response · Authors · 2023-11-17
> **Response to Reviewer h68g**
>
> Thank you so much for taking the time to read this paper.

---

### Official Review · Reviewer_acuW · 2023-11-01

**Soundness:** 3 good
**Presentation:** 2 fair
**Contribution:** 3 good
**Rating:** 6
**Confidence:** 2

**Summary:**

This paper investigates the certified robustness offered by randomized smoothing. Specifically, it addresses the curse of dimensionality associated with standard randomized smoothing algorithms. The authors introduce the concept of dual randomized smoothing, which partitions the input into two subsamples with dimensions $m$ and $n$ such that $m+n=d$. Each subsample is processed by a randomized smoothing classifier. The final decision of the algorithm is based on the most probable class from both classifiers. As a consequence, the dual randomized smoothing enhances the dimension dependence of $\ell_2$ robustness radius from $O(1/\sqrt{d})$ to $O(1/\sqrt{m} + 1/\sqrt{n})$.

**Strengths:**

- This paper addresses the curse of dimensionality of DS and improves dimension dependence from $O(1/\\sqrt{d})$ to $O(1/\\sqrt{m}+1/\\sqrt{n})$. The idea of partitioning input into subsamples into subsamples is very interesting.

- The proposed algorithm is easy and efficient to implement. Also, experiments show DRS have improved empirical performance over standard DS.

**Weaknesses:**

- The improvement from $O(1/\\sqrt{d})$ to $O(1/\\sqrt{m}+1/\\sqrt{n})$ is mostly a constant change.

- Part of DRS in Section 4.2 a bit unclear. Specifically, how is the downsampling function implemented, and how does it make sure that every subsample gets enough spatial info from the original input?

**Questions:**

(Based on first point in weakness) Have you considered further partitioning the input, for instance, into $k$ subsamples, and then leveraging $k$ randomized smoothing classifiers for the final decision? Would such an approach potentially enhance the dimension dependence even more, perhaps improving the robustness radius from $O(1/\sqrt{d})$ to something like $O(\sum_{i=1}^k 1/\sqrt{d_k})$, where $\sum_{i=1}^k d_k = d$?

---

> ### Author Response · Authors · 2023-11-17
> **Response to Reviewer acuW**
>
> Thank you so much for taking the time to read this paper and giving constructive feedback. Please find our response below.
>
> > #### **W1: The improvement from ${O}(1/\sqrt{d})$ to ${O}(1/\sqrt{m}+1/\sqrt{n})$ is mostly a constant change. Would $k$-partitioning based smoothing potentially enhance the robustness radius from ${O}(1/\sqrt{d})$ to something like ${O}(\sum\nolimits_{i = 1}^k {1/\sqrt{d_i}})$, where $\sum\nolimits_{i = 1}^k{1/\sqrt{d_i}}=d$.**
>
> For $m=n=\frac{d}{2}$ implemented in DRS, the improvement of the upper bound is a constant factor of $\sqrt 2$. Reflecting on your discerning insight, in Appendix A.3 of the revised manuscript, we further explore the feasibility of k-partitioning based smoothing. By deriving a general architecture of the smoothed classifier $g$ using $k$-partitioning, in Equation (26), we reformulate the guarantee of certified robustness as an optimization problem. Our findings in Appendix A.3 indicate that:
>
> 1. for $k=2$,  this optimization is proven to be convex, where the global minimum to guarantee the certified robustness is achieved by $\mathop{p'}\nolimits_A^1=\mathop{p'}\nolimits_A^2=\frac{1}{2}\tilde p$. This is in alignment with our assertion in Equation (11).
> 2. For $k>2$, this optimization is non-convex. Providing a numerically stable solution for the global minimum under these circumstances is challenging (the minimum might not exist). A saddle point is achieved when $\mathop{p'}\nolimits_A^j  = {{\tilde p} \mathord{\left/{\vphantom {{\tilde p} k}} \right.} k} , \forall j \in \{1, \ldots, k\}$, but it does not strictly ensure the certified robustness.
>
> We present an in-depth analysis and an illustrative visualization of this $k$-partitioning based smoothing in Appendix A.3, which may help future research endeavors.
>
> > #### **W2: Part of DRS in Section 4.2 is a bit unclear. Specifically, how is the down-sampling function implemented, and how does it make sure that every sub-sample gets enough spatial information from the original input?**
>
> In our revised paper, we added the explanation that: "Considering the high information redundancy of adjacent pixels in the image, we utilize two $2\times2$ pixel-index matrices as the down-sampling kernels. Each kernel selectively retains pixels along one of the two diagonals within the $2\times2$ image block, as depicted in Figure 2. The two sub-images are generated by sliding the two kernels across the original image, moving with a stride of 2."

---

> > ### Comment · Reviewer_acuW · 2023-11-18
> > **Response to Rebuttal**
> >
> > Thank you for the detailed responses and the updated revisions. The revised version has improved presentation. Also it's interesting to see that the problem becomes significantly more challenging when $k\ge 2$.  However, I still believe the dimension improvement by a factor of $\sqrt{2}$ is a limitation of this work, and I remain my previous rating.

---

> > > ### Author Response · Authors · 2023-11-19
> > > **Response to Reviewer acuW**
> > >
> > > Dear Reviewer acuW,
> > >
> > > Thank you so much for your positive and constructive feedback, which is very helpful and makes our paper stronger! We are always available and eager to address any additional questions you might have during our discussion.
> > >
> > > Best regards,
> > >
> > > Authors of Paper1829

---

### Author Response · Authors · 2023-11-17
**Summary of Modification**

We express our sincere appreciation to all the reviewers for their elaborate and constructive feedback. We highlight the changes in blue color in the updated manuscript and summarize the modifications as follows:

1. We presented the analysis of k-partitioning based smoothing in Appendix A.3.
2. We presented the analysis of smoothing with different variances in Appendix A.4.
3. We gave the justification for the tightness of DRS robustness bound in Appendix A.5.
4. We discussed two more relevant works mentioned by Reviewer iAVM in Section 2.
5. We explained the parameter $5*10^{-7}$ in Section 3.2.
6. We explained the implementation of the down-sampling algorithm in Section 4.2.
7. We added a clarification of the pre-trained model for the base classifier in Section 5.1.
8. We capitalized the initial letter in Section 5.3.

---

### Meta-Review · Area_Chair_csFd · 2023-12-15

**Metareview:**

This paper studies randomized smoothing by providing a certified robustness for high-dimensional input via the utilization of dual smoothing in a lower-dimensional space. Two reviewers are very appreciative of the contributions of the paper. Two other reviewers have acknowledged the novelty of the work but still have a few concerns. For example, the constant improvement by the paper may not be significant enough. The authors’ responses largely address the concerns.

**Justification For Why Not Higher Score:**

I reached this decision by evaluating the contributions and novelty of the work, taking into consideration both the reviews and the responses from the authors.

**Justification For Why Not Lower Score:**

I reached this decision by evaluating the contributions and novelty of the work, taking into consideration both the reviews and the responses from the authors.

---

### Decision · Program_Chairs · 2024-01-16

Accept (poster)